# Neural Signals Associated with Orienting Response and Arousal Inhibition in Concealed Information Test

**DOI:** 10.3390/bs14080627

**Published:** 2024-07-23

**Authors:** Wang Feng, Fei Wang, Hongyi Zhu, Chen Jiang, Liyang Sai

**Affiliations:** 1Department of Psychology, Hangzhou Normal University, Hangzhou 311121, China; 2022111004052@stu.hznu.edu.cn (W.F.);; 2Zhejiang Philosophy and Social Science, Laboratory for Research in Early Development and Childcare, Hangzhou Normal University, Hangzhou 311121, China; 3Zhejiang Key Laboratory for Research in Assessment of Cognitive Impairments, Hangzhou Normal University, Hangzhou 311121, China

**Keywords:** concealed information test, orienting response, arousal inhibition, ERPs

## Abstract

Recent theory suggests that both the orienting response and arousal inhibition play roles in the effect of the concealed information test (CIT). However, the neural signatures associated with these two processes remain unclear. To address this issue, participants were motivated to either conceal or reveal crime-related stimulus during CIT while EEG was recorded. By using a temporal principal component analysis, we found that crime-related stimuli produced a larger early P3 than crime-irrelevant stimuli in both the conceal condition and reveal condition. This result suggests that this early P3 reflects an orienting response. In addition, we found that crime-related stimuli elicited a larger frontal negative slow wave than crime-irrelevant stimuli in the conceal condition but not the reveal condition, which suggests that the frontal negative slow wave reflects the arousal inhibition process. These results provide crucial evidence for understanding the neural basis underlying CIT.

## 1. Introduction

The concealed information test (CIT) is widely used to detect information that individuals are familiar with but are trying to conceal. In a typical CIT, a crime-related stimulus (also called a probe) is presented among a series of crime-irrelevant stimuli (irrelevant), and participants are asked to respond whether they know these items. A number of studies have found that the probe evoked unique physiological responses such as greater skin conductance response [1], shorter respiration line length [2] and slower heart rate [3] than irrelevant. The orienting response (OR) theory has been proposed to explain this CIT effect [4]. The OR is described as a complex of behavioral and physiological reactions in response to any novel stimuli [5]. Importantly, when a stimulus carries a special significance to individuals, an enhanced orienting response will occur [6]. For participants who had knowledge of a mock crime, the probe is distinguishable from irrelevant stimuli and holds significance, evoking unique physiological responses. In contrast, for participants who did not have knowledge of a mock crime, the probe resembles irrelevant stimuli and lacks significance, evoking physiological responses similar to the irrelevant stimuli.

Recently, some researchers have proposed that attempting to inhibit physiological arousal also contributes to the observed CIT effect (the arousal inhibition theory). Specifically, the arousal inhibition theory suggests that the guilty participants would try to suppress their physiological responses to avoid being detected during CIT, which leads to the changes in physiological responses [7]. Klein Selle and her colleagues conducted a series of studies to examine the role of orienting response and arousal inhibition in the observed CIT effect during the CIT based on the autonomic nervous system (ANS) [8,9,10]. Specifically, two conditions were included in these studies: the conceal condition, in which participants were asked to inhibit their arousal to the probe, and the reveal condition, in which participants were asked to reveal their arousal to the probe. Results showed that the probe was associated with the increased SCR compared with the irrelevant in both the conceal condition and the reveal condition, while the probe was associated with heart rate deceleration and respiratory suppression compared with the irrelevant in the conceal condition only, not in the reveal condition. The authors argued that the increased SCR observed in both conditions reflected participants’ orienting response to the probe, while the decreased RLL and HR observed in the conceal condition reflected participants’ efforts to inhibit their arousal to the probe [10].

Although the above research provided evidence about two different mechanisms underlying CIT and their associated autonomic physiological measures, their associated neural signatures were still unclear. To date, four studies have tried to examine this issue by using the EPR technique [11,12,13,14]. All four studies used paradigms similar to klein Selle et al., in which participants were asked to inhibit their arousal to the probe in the conceal condition and were asked to reveal their arousal to the probe in the reveal condition. P3 has been proposed to reflect the orienting response like SCR. P3 is a positive ERP component that typically appears 300–800 ms after stimuli, and it has been found to be sensitive to stimuli that are significant and meaningful to participants [15,16,17]. The existing findings are not consistent. For example, Kubo and Nittono [11] found that the P3 effect was larger for the reveal condition than the conceal condition (also see [13], but Rosenfeld et al. [12] found that the P3 effect was larger for the conceal condition than the reveal condition. Klein Selle et al. [14] found a similar P3 effect for both the conceal condition and the reveal condition.

Two ERP components have been proposed to reflect arousal inhibition during CIT. One of them is N2. N2 is a negative ERP component that typically occurs 250–350 ms after stimuli, and it has been found to be associated with cognitive control such as conflict detection and monitoring [18]. Rosenfeld et al. [12] examined the role of arousal inhibition on N2, and they did not find the probe was associated with an increased N2 amplitude compared with the irrelevant in the conceal condition, suggesting N2 is not the indicator of inhibition arousal during CIT (also see [13,14]).

The other ERP component is a frontal negative slow wave (FNSW). The FNSW typically occurs around 500 to 1000 ms after presenting the stimulus and originates from the frontal cortex [19]. It has been suggested to be linked to cognitive load [20,21]. Matsuda and Nittono [13] reported that the probe elicited a larger right than left frontal slow wave than the irrelevant in the conceal condition, but this effect was not found in the reveal condition. Thus, they argued that the frontal negative slow wave reflected the arousal inhibition process during CIT.

Given the above limited and controversial findings, the present study aims to use the ERP technique to explore neural signatures associated with orienting response and arousal inhibition again. In addition, we expanded the above studies in two ways. Firstly, because the time window of N2 and P3 and the time window of P3 and FNSW are overlapped, we used temporal principal component analysis (PCA) to score them. Secondly, the interval between stimuli in the previous research was short (1300 ms used by klein Selle et al. [14]) so that participants might not have enough time to inhibit their arousal state [22], which might be why they did not observe ERP components associated with arousal inhibition. In this study, we prolonged the interval between the stimuli to 2500–2800 ms to test arousal-related and inhibition-related ERP components. We used the design by klein Selle et al. [8]. Specifically, there were two conditions in the study: the conceal condition and the reveal condition. In the conceal condition, participants were asked to perform a mock crime and then motivated to inhibit their physiological arousal during CIT. In contrast, participants in the reveal condition were instructed to watch a video about a mock crime and then instructed to allow their natural physiological arousal to occur during CIT.

Based on previous findings, we expected that the probe would produce larger P3 amplitude than the irrelevant in both the conceal condition and the reveal condition. However, it is hard to make a prediction about whether there is a significant difference between the conceal condition and the reveal condition because of previous controversial findings [11,12,13,14]. Second, given that prior research did not find an increased N2 effect in the conceal condition [12,13,14,23], we did not expect that the probe would produce a more negative N2 amplitude than the irrelevant in the conceal condition. Lastly, because previous research had reported that frontal negative slow wave was associated with arousal inhibition in CIT [13,19], we expected that the probe would evoke a more negative frontal negative slow wave than the irrelevant in the conceal condition but not in the reveal condition.

## 2. Materials and Methods

### 2.1. Participants

A power analysis was performed using G*Power (Version 3.1.9.4, [24]) to determine the necessary sample size. To detect a medium effect size of *f* = 0.25 with a power of 0.95 and a significance level of 0.05, two-tailed, a total sample size of 36 participants (18 participants for each group) for an ANOVA F-test with repeated measures within–between interaction was needed. We recruited 47 participants from Hangzhou Normal University in Hangzhou, China (M_age_ = 21.20 years, 14 male, SD = 2.46, range from 18 to 27 years). Six participants were excluded from analysis due to excessive electroencephalographic (EEG) artifacts. One additional participant was excluded because his data were not saved. Data from 40 participants were analyzed (12 male, M = 20.83, SD = 2.23, range from 18 to 26 years), of whom 20 were in the reveal condition and the other 20 were in the conceal condition. All participants received CNY 50 (about USD 6.89) for their participation. This study was approved by the ethics committee of Hangzhou Normal University, and all participants signed written informed consent forms before participating in the experiment.

### 2.2. Procedure

After entering the laboratory, participants were randomly assigned to either the conceal or the reveal condition. Participants in the conceal condition were asked to commit a mock crime. They were instructed to sneak into an office and steal an item (a pair of earrings) from a safe box. Participants in the reveal condition were instructed to imagine themself as a witness to a case of theft. They viewed a video that showed a criminal sneak into an office and steal an item (a pair of earrings) from a safe box. After that, participants were asked one check questions: “What item did you steal from the office?” (the conceal condition) or “What did the thief steal from the office?” (the reveal condition). If the participants failed to provide accurate answers, they were called upon to return to the office to make sure that they could answer the questions correctly. Afterward, all participants were informed that they would take a lie detection test.

There were six types of items in the CIT. The earring was the probe stimulus, the wallet was the target stimulus, and the four irrelevant stimuli included ring, necklace, watch, and bracelet.

During the CIT, three kinds of stimuli were randomly presented to participants. Participants were instructed to press the “Space” key when they saw the wallet and not to press the key when they saw the other items. Participants in both conditions were reminded about their assigned roles (criminal vs. witness) and informed of the basic principles of the CIT. More specifically, all participants were informed that their physiological responses would change when they recognized an item was related to the crime, such as intensified breathing and accelerated heartbeat. Afterwards, participants in the two conditions were given distinct instructions. The participants in the conceal condition were told that they needed to suppress their physiological responses to the stolen items during the test to avoid being detected as the criminal by the lie detection. The participants in the reveal condition were told that they should not try to suppress their physiological responses to the stolen items, but respond naturally or automatically, to avoid being misjudged by the lie detection as the criminal. All participants were informed that they would receive 20 CNY (about USD 2.76) as a reward if the lie detection did not identify them as the criminal.

Participants sat approximately 1 m in front of the computer and were instructed to place their right index finger on the “Space” key. Each stimulus was presented in white font on a black background. First of all, the screen presented a fixation point of 500–800 ms. Subsequently, a stimulus was displayed in the center of the screen for 2000 ms (see Figure 1). Participants were asked to press the key quickly and accurately when they saw the target stimulus. Each stimulus was repeated 40 times. In sum, there existed 6 × 40 = 240 trials. Following every completion of 60 trials (about 2 min), participants were permitted to rest for a period of time. The entire experiment lasted approximately 12–15 min.

After the CIT, participants were asked to rate the effort they exerted in inhibiting their arousal. They were asked: “Please rate your level of effort in inhibiting physiological responses on a scale from 1 to 7, where 1 represents minimal effort and 7 represents maximal effort”.

### 2.3. EEG Data Acquisition

In this study, we utilized 32 pre-amplified Ag/AgCl electrodes and antiCAP system (Brain Products, Berlin, Germany) to collect the participants’ electroencephalogram signals by placing the electrodes on the elastic electroencephalogram cap following the 10–20 system standard. The reference electrode was placed on the left mastoid, and the average signal from the mastoid electrodes on both ears was recorded for reference. Electrode impedances were maintained below 10 KΩ. The sampling rate was set at 1000 Hz.

A Brain-Vision Analyzer (Brain Products, Berlin, Germany) was utilized for conducting data analyses offline. The ocular correction independent component analysis (ICA) was employed to eliminate artifacts associated with eye movement. Due to the lack of extra horizontal and vertical ocular electrodes, we selected Fp2 as the vertical eye electrode. Electrooculograms were filtered out in a semi-automatic mode and marked, with the amplitude change exceeding the average value or the voltage change value of the amplitude exceeding 97% considered as electrooculogram. Then, we removed channels with the artifact rate greater than 20% through raw data inspection. Continuous EEG data were bandpass filtered from 0.1 to 30 Hz following the procedure of previous studies [12,13]. Continuous EEG data were then segmented into epochs of 2000 ms duration, including a 200 ms pre-stimulus baseline, and time-locked to the onset of the probe or irrelevant stimulus. The extended segment duration aimed to analyze the slow wave, as slow deflections in the EEG last for at least a couple of hundred milliseconds, but they may also extend up to several seconds [21]. Epochs were baseline-corrected, and trials with signals exceeding ±100 μV were defined as artifact trials and excluded. After correcting for eye movements and removing artifacts, for participants in the conceal condition, the mean number of valid trials for probe was 34.45 (SE = 1.29) segments, while for irrelevant stimuli, it was 136.75 (SE = 5.31) segments. For participants in the reveal condition, the mean number of valid trials for probe was 34.10 (SE = 1.19) segments, and for irrelevant stimuli, it was 139.1. (SE = 4.72) segments. ERPs in response to irrelevant stimuli were averaged across all four irrelevant items.

Our analyses focused on N2, P3, and FSNW as we hypothesized. Given these ERPs are largely overlapped, a temporal PCA was conducted to analyze them by using ERP-PCA toolkit version 2.86 [25]. Specifically, the temporal PCA was conducted by combining data from the conceal and reveal conditions and using the Promax rotation method with a kappa of 3, which is the parameter that determines how oblique the rotations will be [26]. A total of 2000 time points of the average ERP of each participant were used as variables (200 ms before the start of stimulus as a baseline), with participants and conditions as observations. According to the scree plots (See the screen diagram of time PCA in the Appendix A), a total of 26 factors could be extracted, and 14 factors accounted for more than 1% of variance and were retained for further examination (see Table 1).

### 2.4. Statistical Analysis

Group analyses were conducted using ANOVA methods and *t*-tests, with the effect size of significant findings expressed through partial eta squared and Cohen’s d. Additionally, Jeffreys–Zellner–Siow (JZS) Bayes factors (BFs) (see Rouder et al. [27]) were employed to supplement traditional statistical inference. The BF is a numerical value that serves as a means of quantifying the ratio of the probability of the null hypothesis to the probability of the alternative hypothesis [28]. For all *t*-tests, either the BF_10_ (favoring the alternative hypothesis) or the BF_01_ (favoring the null hypothesis) was reported, while for ANOVA effects, either the *BF_Inclusion_* (favoring the alternative hypothesis) or *BF_Exclusion_* (favoring the null hypothesis) was reported, reflecting a comparison of all models containing a particular effect to those without the effect (also see klein Selle et al. [14]). A BF value of ≥3 was regarded as moderate evidence for the respective hypothesis [28]. BFs were computed using JASP (Version 0.14.1).

## 3. Results

Based on the temporal and spatial characteristics of P3, two factors were chosen for analysis (We conducted temporal PCA on all 32 electrodes and plotted the grand mean (*N* = 40) ERP waveforms of the analyzed components (TF01, TF02, TF03, TF04) (−200 to 2000 ms, 200 ms pre-stimulus baseline) at 32 electrodes for the probe and irrelevant items. However, given that the figures are large, we think it is not appropriate to put them in the main text. We decided to put the figures in the Appendix A). These factors included a positivity peaking at 400ms at P, and a positivity peaking at 590 ms at Pz. We did not find any components that were temporally and spatially similar to N2 or FNSW. Nonetheless, we observed a negativity peak at 1022 ms at Fz and a negativity peak at 1850 ms at Fz. These were examined as exploratory analyses.

### 3.1. P300 to Probe vs. Irrelevant (The Parietal Positivity Peaking at 400 ms)

A 2 (stimulus type: probe vs. irrelevant) by 2 (condition: conceal vs. reveal) mixed ANOVA with the 400 positivity at Pz showed a significant main effect of stimulus type *F* (1, 38) = 10.25, *p* = 0.003, *η*^2^ = 0.06, *BF_inclusion_* = 10.47. Simple contrasts revealed the amplitude of the positivity elicited by the probe was larger than that elicited by the irrelevant (2.49 ± 0.65 μV vs. 0.70 ± 0.47 μV). In addition, there was a significant interaction between stimulus type and condition, *F* (1, 38) = 4.48, *p* = 0.04, *η*^2^ = 0.03, but this result was not supported by the *BF_inclusion_* = 1.78. The main effect of conditions was not significant, *F* (1, 38) = 3.04 × 10^−5^, *p* = 0.99, *η*^2^ = 0.01, *BF_inclusion_* = 0.21.

### 3.2. P300 to Probe vs. Irrelevant (The Parietal Positivity Peaking at 590 ms)

A 2 (condition: conceal vs. reveal) by 2 (stimulus type: probe vs. irrelevant) mixed ANOVA was conducted with the positivity at 590 ms at Pz as the dependent variable. Results showed a significant main effect of stimulus type, *F* (1, 38) = 43.44, *p* < 0.001, *η*^2^ = 0.22, *BF_inclusion_* = 4.82 × 10^4^. Simple contrasts revealed that the amplitudes of the P3 elicited by the probe were larger than those elicited by the irrelevant (4.86 ± 0.73 μV vs. 0.29 ± 0.47 μV). Meanwhile, there was a significant main effect of condition, *F* (1, 38) = 6.61, *p* = 0.01, *η*^2^ = 0.07, *BF_inclusion_* = 3.5, revealing that P3 amplitudes in the reveal condition were larger than those in the conceal condition (3.89 ± 0.72 μV vs. 1.27 ± 0.72 μV). There was a significant interaction between stimulus types and condition, *F* (1, 38) = 16.62, *p* < 0.001, *η*^2^ = 0.09, *BF_inclusion_* = 142.85. Follow-up tests revealed that the probe induced a more positive amplitude compared with irrelevant in both the conceal condition (2.14 ± 0.72 μV vs. 0.40 ± 0.69 μV, *t* (19) = 2.9, *p* = 0.01, *d* = 0.65, 95% CI = [0.16, 1.13], *BF*_10_ = 5.50) and the reveal condition (7.59 ± 1.27 μV vs. 0.19 ± 0.65 μV, *t* (19) = 7.54, *p* < 0.001, *d* = 1.32, 95% CI = [0.71, 1.92], *BF*_10_ = 2.08 × 10^3^), and the probe minus irrelevant differences in the reveal condition was greater than that in the conceal condition (7.40 ± 1.25 μV vs. 1.74 ± 0.60 μV, *t* (19) = 4.00, *p* < 0.001, *d* = 0.90, 95% CI = [0.37, 1.41], *BF*_10_ = 46.95). (for ERP waveforms and topographies, see Figure 2, Figure 3, Figure 4 and Figure 5).

### 3.3. Exploratory Analyses

#### 3.3.1. The Frontal Negativity Peaking at 1022 ms

The same ANOVA was conducted on the 1022 negativity at Fz. Results showed a significant main effect of stimulus type, *F* (1, 38) = 4.59, *p* = 0.04, *η*^2^ = 0.04, suggesting the negativity elicited by the probe was larger than that elicited by the irrelevant (−1.54 ± 0.57 μV vs. −0.34 ± 0.38 μV), but this result was not supported by the *BF_inclusion_* = 2.03. There was no other significant main effect or interaction (ps > 0.1).

#### 3.3.2. The Frontal Negativity Peaking at 1850 ms

The same ANOVA was conducted on the 1850 negativity at Fz. Results showed that the main effect of the condition was significant, *F* (1, 38) = 5.14, *p* = 0.03, *η*^2^ = 0.06, suggesting the negativity in the conceal condition was more negative than that in the reveal condition (−0.41 ± 0.46 μV vs. 1.08 ± 0.46 μV). However, this result was not supported by the *BF_inclusion_* = 1.49. The main effect of stimulus type was not significant, *F* (1, 38) = 3.32, *p* = 0.08, *η*^2^ = 0.03, *BF_inclusion_* = 1.06. The interaction between the condition and stimulus type was significant, *F* (1, 38) = 8.46, *p* = 0.006, *η*^2^ = 0.08, *BF_inclusion_* = 13.16. Follow-up analyses showed that in the conceal condition, the probe evoked a more negative wave than the irrelevant (−1.83 ± 0.51 μV vs. 1.02 ± 0.38 μV, *t* (19) = −4.46, *p* < 0.001, *d* = −1.00, 95% CI = [−1.53, −0.45], *BF*_10_ = 114.81). There was no significant difference between probe and irrelevant in the reveal condition (1.41 ± 0.96 μV vs. 0.74 ± 0.52 μV, *t* (19) = 0.65, *p* = 0.52, *d* = 0.15, 95% CI = [−0.30, 0.59], *BF*_10_ = 0.28).

#### 3.3.3. Was the Negativity at 1850 ms Related to Arousal Inhibition?

Given that we found the probe elicited a larger negativity at 1850 at Fz than the irrelevant only in the conceal condition, but not in the reveal condition, it is possible that the negativity at 1850 ms was associated with arousal inhibition. To validate this hypothesis, we initially calculated the difference in amplitudes between the probe and irrelevant and then performed a correlation analysis between the difference amplitudes and self-reported inhibition arousal. The result indicated a significant negative correlation between the difference in frontal negativity at 1850 ms and the level of effort participants exerted to suppress their arousal (*r* = −0.37, *p* = 0.02, *BF*_10_ = 5.41; see Figure 6).

## 4. Discussion

Based on the response fractionation theory, the present study used the ERPs technique to examine neural signatures associated with orienting response and arousal inhibition in an ERP-based CIT. Results showed that the probe elicited larger positivity at both 400 ms and 590 ms than the irrelevant in both the reveal condition and the conceal condition. The positivity at 590 ms was larger in the reveal condition than in the conceal condition, while no significant difference was found in positivity at 400 ms between the two conditions. Exploratory analyses showed that the probe elicited a larger frontal negativity peaking at 1850 ms than the irrelevant in the conceal condition but not the reveal condition. Further correlation analyses confirmed that this negativity was associated with self-reported inhibition scores. These findings provide evidence about the neural correlates of orienting response and arousal inhibition in CIT.

### 4.1. Neural Signatures Associated with Orienting Response

By using temporal PCA, we found that the probe produced two larger positive waves than irrelevant in both the conceal and reveal condition: a positive wave peaking at 400 ms and another positive wave peaking at 590 ms. These temporal characteristics closely resemble those of P3a and P3b [29,30,31,32], suggesting the possibility that they are indeed P3a and P3b. The P3a is a positive ERP component typically occurring around 300-450 ms in visual paradigms [29,30]. Previous studies have indicated the P3a reflects the involuntary brain’s instinctive, passive attention-shifting process to external stimuli or orienting, which does not involve active attention and memory processes [33,34,35]. Because the probe contained special significance for participants in both conditions, the probe automatically attracted participants’ attention when it was presented. Therefore, P3a here may reflect the involuntary shift of participants’ attention toward significant stimuli.

P3b is a positive ERP at the Pz site typically peaking between 400 and 700 ms [31,32]. Our result is consistent with findings from klein Selle et al. [14], which also found that the probe elicited larger P3 than the irrelevant in both the conceal condition and reveal condition, although they did not use PCA to score P3. P3b activity is related to updating operations and subsequent memory storage, reflecting the amount of voluntary attention resources invested during stimulus processing [16,36]. In this study, the P3b is considered to reflect the voluntary attention or recognition process to the probe after the orienting response. However, it should be noted that the P3b effect size was much greater in the reveal condition (*d* = 1.32) than in the conceal condition (*d* = 0.65). One possible explanation for this result is that participants in the conceal condition needed to inhibit their arousal to the probe, which led to them consciously reducing their attention to the probe, resulting in smaller P3b. Our result is consistent with results from two previous CIT studies for examining intention to conceal knowledge, which also found a greater P3 difference in the reveal condition than in the conceal condition [11,19]. However, our results are not consistent with results from Rosenfeld et al. [12]. They found that P3b effect size was greater in the conceal condition than in the reveal condition, and they argued that the probe was more meaningful to the suspects than to the witnesses. One possible reason for these different results could be due to variations in the experimental paradigms used. Unlike the CIT paradigm used in this study, Rosenfeld et al. [12] used the CTP paradigm, which separated the implicit detection task of identifying relevant or irrelevant stimuli from the explicit discrimination task of identifying target or non-target stimuli [37]. Participants in the CTP also needed to perform keystroke responses when detecting the presence of the probe. Because individuals in the conceal condition were told to inhibit their arousal of the probe as much as possible, the probe became the only stimulus conflicting with the key command, which potentially increased the participants’ active attention. On the other hand, for participants in the reveal condition, neither the probe nor irrelevant stimuli conflict with the key command. It should be noted that some previous research suggests that mental effort also modulates P3 amplitudes [17,38]. For example, Koeckritz et al. [17] reported that concealing untrustworthiness showed a smaller parietal mean P3 amplitude compared with truthful response toward untrustworthy faces [17]. Given that arousal inhibition typically requires much mental effort, it is also possible that P3 is also influenced by arousal inhibition.

### 4.2. Neural Signatures Associated with Arousal Inhibition

We expected that two ERP components would be associated with arousal inhibition based on previous research: the N2 and the frontal negative slow wave. In contrast to our prediction, we did not find that the probe produced a more negative N2 than the irrelevant in the conceal condition. Our result was consistent with findings from Rosenfeld et al. [12] and klein Selle et al. [14], which also did not find that the probe produced a more negative N2 than the irrelevant in the conceal condition. However, it should be noted that Rosenfeld et al. [12] reported that the probe was associated with a longer N200-N300 latency than the irrelevant in the conceal condition, but that finding should be interpreted cautiously because it was not an often-reported result. Results from klein Selle et al. [14] and the current study did not replicate that finding.

Unexpectedly, we found that the probe produced a more negative slow wave peaking at 1850 ms at the frontal site than the irrelevant in the conceal condition but not in the reveal condition. Due to the extended segmentation time in this study, along with the implementation of principal component analysis, the latency of frontal slow waves was longer than that reported by Matsuda & Nittono [13]. The negative slow wave is typically found to be associated with cognitive load, with greater cognitive load associated with a more negative frontal slow wave [20,21]. For instance, in studies on mental rotation, slow potential amplitude would be higher for trials with visually complex stimuli compared with trials with simple stimuli [39]. The frontal negative slow wave found in the conceal condition may be because the progress of inhibiting physiological arousal rather than the progress of allowing physiological arousal to occur required more cognitive resources, which was accompanied by a higher cognitive load. This argument is backed by our result indicating a significant negative correlation between the difference in amplitude of FNSW and the scores that participants reported on how much they had inhibited their physiological responses. Our result is similar to the result from Matsuda and Nittono [19], which also found that the probe produced a more negative FNSW than irrelevant items only in the conceal condition, in which participants had to conceal the recognition of the crime-relevant item. However, our results do not entirely match the finding of Matsuda and Nittono [13], as they observed that the frontal negative slow wave amplitude was larger (more negative) for the probe than for the irrelevant in both the conceal and reveal conditions. One possible reason is that participants in the reveal condition were asked to try to reveal their recognition to the probe by increasing brain activities [13], which could have also heightened participants’ cognitive load. However, in the present study, participants were simply instructed to let their automatic and natural physiological responses take place, which did not require cognitive resources.

### 4.3. Limitations

There are several limitations to this study. Firstly, this study examined neural signatures associated with the orienting response and arousal inhibition response during CIT. However, it is unclear how the central neural system and peripheral responses (SCR or HR) interact with each other during these two responses. Further studies could concurrently record the ERP components and peripheral responses to investigate this issue. Additionally, while we have demonstrated neural signatures associated with orienting response and arousal inhibition, it remains uncertain how effectively these neural signatures could be employed to detect concealed information. Further studies could involve an innocent group to explore this issue.

## 5. Conclusions

In sum, this study was designed to explore the neural signatures associated with orienting response and arousal inhibition in CIT. Our results revealed that the probe elicited a more positive P3a compared with the irrelevant in both the conceal and reveal conditions, suggesting a potential association between P3a and the orienting response. In addition, our exploratory results showed that the probe induced a larger frontal negative slow wave compared with the irrelevant in the conceal condition but not the reveal condition, indicating that the frontal negative slow wave was associated with the arousal inhibition process. These findings provide important evidence for the neural basis of the response fractionation theory.

## Figures and Tables

**Figure 1 behavsci-14-00627-f001:**
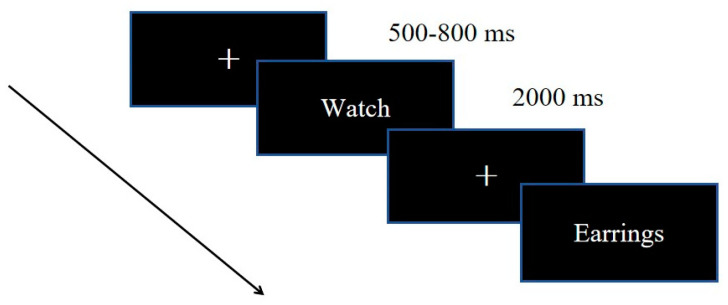
The task structure of the concealed information test.

**Figure 2 behavsci-14-00627-f002:**
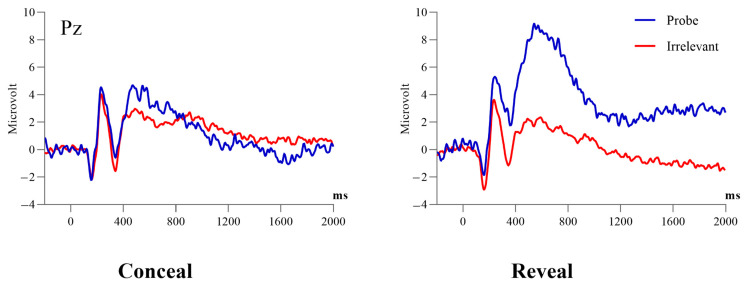
Grand-average ERP waveforms evoked by the probe and irrelevant as measured at the Pz electrode.

**Figure 3 behavsci-14-00627-f003:**
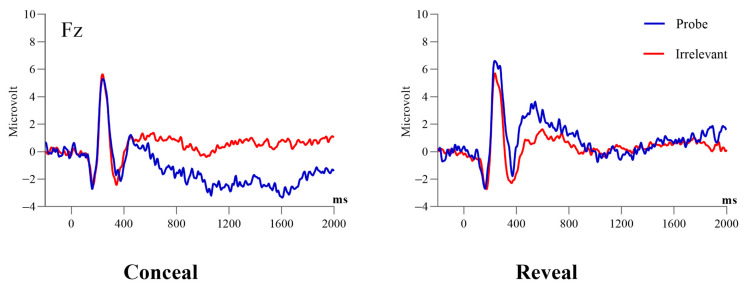
Grand-average ERP waveforms evoked by the probe and irrelevant as measured at the Fz electrode.

**Figure 4 behavsci-14-00627-f004:**
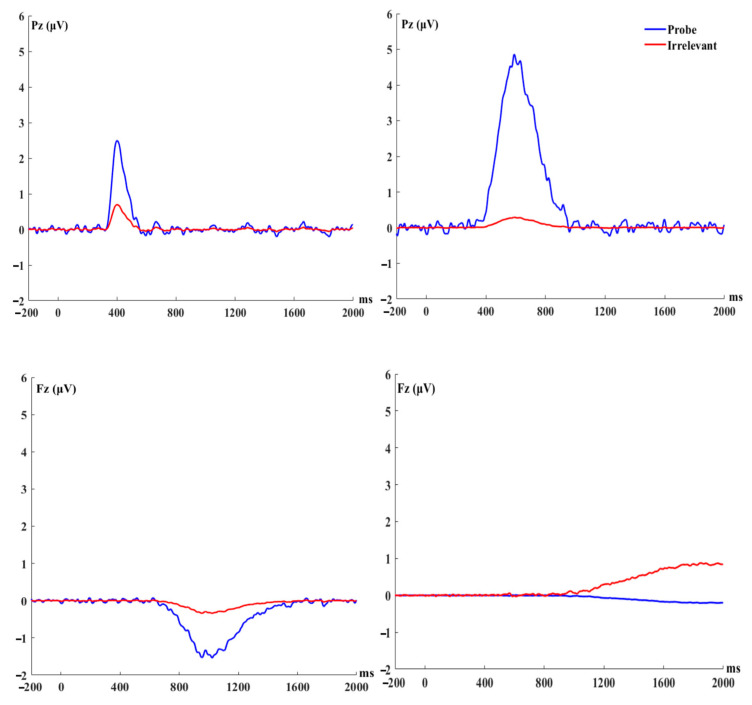
Virtual ERP components extracted from the temporal PCA (combining conceal and reveal conditions).

**Figure 5 behavsci-14-00627-f005:**
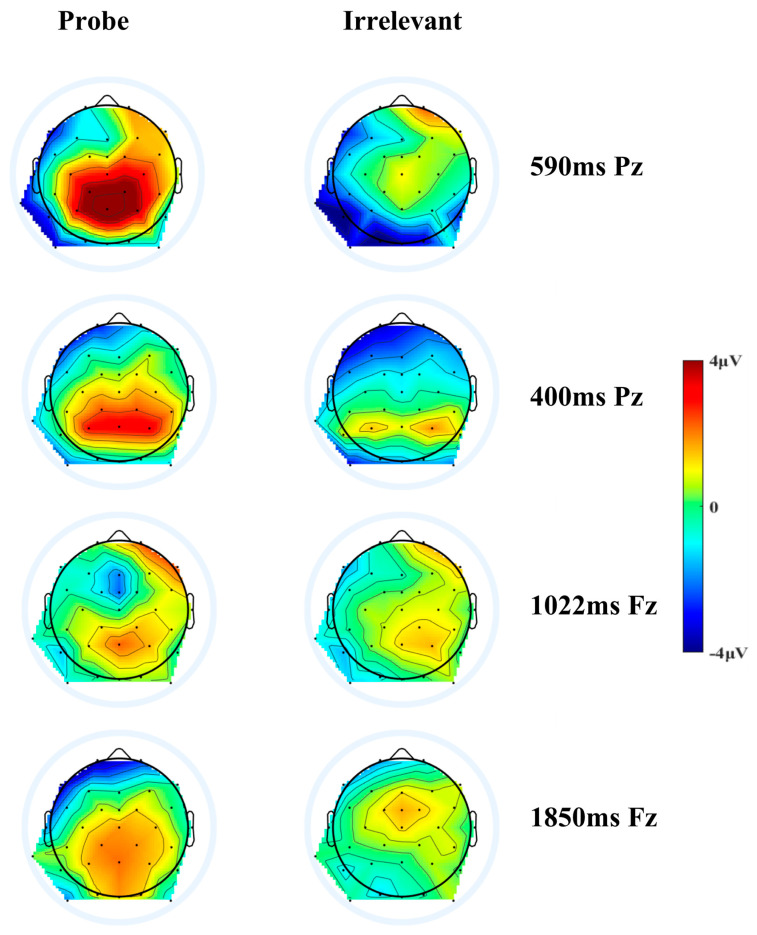
Scalp topographies of temporal principal component analyzed ERPs locked to probe or irrelevant.

**Figure 6 behavsci-14-00627-f006:**
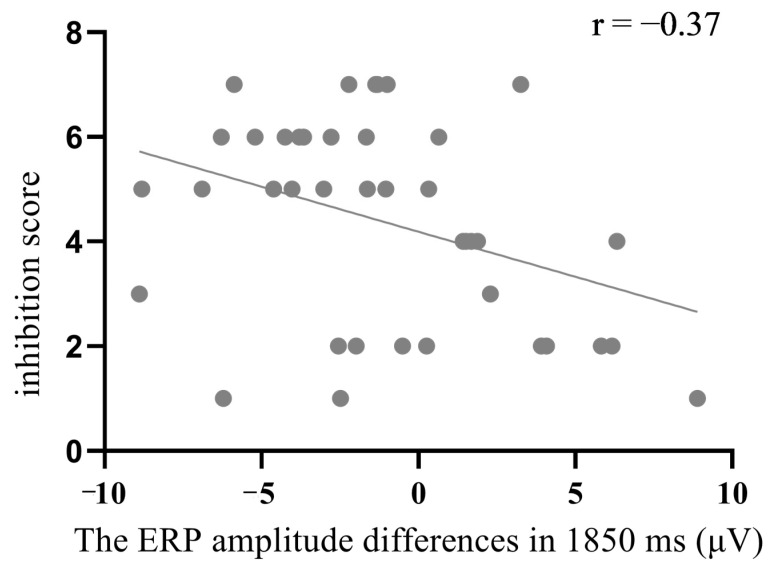
The scatter plots: the ERP amplitude differences in 1850 ms at the frontal region vs. inhibition score.

**Table 1 behavsci-14-00627-t001:** Description and analysis of variance results for each temporal factor.

Temporal Factors	Peak Loading (ms)	Variance (%)	Polarity
TF01	1850	25.3	−
TF02	590	17.9	+
TF03	1022	17.3	−
TF04	400	6.2	+

## Data Availability

The data presented in this study are available on request from the corresponding author. The data are not publicly available due to privacy restrictions.

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
