# Peer review of "Neural Signals Associated with Orienting Response and Arousal Inhibition in Concealed Information Test"

_behavsci, 2024, doi:10.3390/bs14080627_

Round 1
Reviewer 1 Report
Comments and Suggestions for Authors
Review of Manuscript: Neural Signals Associated with Orienting Response and Arousal Inhibition in Concealed Information Test
This study investigates neural indicators of orienting response and arousal inhibition using ERP techniques and a modified task featuring two conditions: conceal and reveal.
Authors hypothesized a larger P3 amplitude in response to probes compared to irrelevant stimuli in both conditions, despite existing controversy regarding differences between the two conditions.
Methods:
While the sample size was calculated using G*Power, the n = 20 per group appears small. A larger sample size, perhaps 36 participants per group, may yield more robust results.
The conceal and reveal conditions differ not only in the act of concealing or revealing answers (physiologically or on the scale measuring effort to conceal physiological responses) but also in the role in the crime; one group commits the act while the other serves as a witness. Having the same role, with the only difference being concealment or revelation, would help reconcile discrepancies between findings in both conditions.
Physiological response measures should have been included to confirm participants' adherence to experimental conditions.
Results:
Significant results were only found for P3 amplitude in response to probes versus irrelevant stimuli (at 400 ms and 590 ms). Differences were observed between probes versus irrelevant stimuli and between the reveal and conceal conditions, with amplitudes being larger in the reveal condition. An interaction between condition and stimulus type was also noted.
Figures (2–4) could be improved by indicating statistical significance and the time window of the analysis. Font size in Figure 4 is too small, making it difficult to read.
The description of Figure 6 in the text contradicts the presentation in the graph; while the text suggests a positive relationship, the slope in the figure indicates a negative one.
Limitations include the absence of peripheral physiological response measures, which could have confirmed differences between conditions as instructed.
Author Response
Reviewer 1
Comment 1: While the sample size was calculated using G*Power, the n = 20 per group appears small. A larger sample size, perhaps 36 participants per group, may yield more robust results.
Response: Thank you for this comment. We agree with the reviewer that a larger sample size may yield more robust results. We also provide BF to show whether our results are robust. As you can see from results, BF of our main results are typically >3, which indicates relative strong evidence supporting our results.
Comment 2: The conceal and reveal conditions differ not only in the act of concealing or revealing answers (physiologically or on the scale measuring effort to conceal physiological responses) but also in the role in the crime; one group commits the act while the other serves as a witness. Having the same role, with the only difference being concealment or revelation, would help reconcile discrepancies between findings in both conditions.
Response: Thanks for this comment. We used the same design and procedure as previous research (see, Klein et al. 2016, 2017, 2019, 2021; Rosenfeld et al., 2017).We agree with the reviewer that having same role would have a better control for the two conditions. There are two reasons for current design: (1) asking guilty participants to enhance their arousal seems not to be reasonable; (2) Keeping the design consistent with previous research will make our research comparable to previous research.
Comment 3: Physiological response measures should have been included to confirm participants' adherence to experimental conditions.
Response: Thanks for this valuable suggestion. Considering the time length for measuring physiological responses such as skin conductance, respiration, and heart rate (14-19s) is much longer than that for measuring EEG signals (1-2s), we did not conduct simultaneous measurements of these. However, we asked participants to make self-report about how much effort they use to inhibit their arousal after the main experiment. The results showed that participants in the conceal condition exerted more effort in suppressing physiological arousal compared to those in the reveal condition (5.95±0.19 vs. 2.80±0.31, t (38) = 8.67, p < .001, d = 0.53, 95% CI = [2.42, 3.89], BF10 = 4.13×107).
Comment 4:Figures (2–4) could be improved by indicating statistical significance and the time window of the analysis. Font size in Figure 4 is too small, making it difficult to read.
Response: Thanks for this comment. We used PCA analyses to score ERPs rather than using mean amplitude of certain time windows so that it is impossible to show the time window and statistical significance of analyzed ERPs in the original ERP waves such as Figure 2 and Figure 3. In addition, we have revised Figure 4 to make it easy to read.
Comment 5:The description of Figure 6 in the text contradicts the presentation in the graph; while the text suggests a positive relationship, the slope in the figure indicates a negative one.
Response: We apologized for this mistake. We have revised it now.
"The result indicated a significant negative correlation between the difference in frontal negativity at 1850ms and the level of effort participants exerted to suppress their arousal (r = -0.37, p = 0.02, BF10 = 5.41; see Figure 6)"see page 10
References
klein Selle, N., Agari, N., & Ben-Shakhar, G. (2019). Hide or Seek? Physiological Responses Reflect Both the Decision and the Attempt to Conceal Information. Psychological Science, 30(10), 1424–1433. https://doi.org/10.1177/0956797619864598
klein Selle, N., Gueta, C., Harpaz, Y., Deouell, L. Y., & Ben-Shakhar, G. (2021). Brain-based concealed memory detection is driven mainly by orientation to salient items. Cortex, 136, 41–55. https://doi.org/10.1016/j.cortex.2020.12.010
Klein Selle,N.,Verschuere,B.,Kindt,M.,Meijer,E.和Ben-Shakhar,G.(2016)。隐藏信息测试中的定向与抑制:不同的认知过程驱动不同的生理测量:CIT 中的定向和抑制过程。 心理生理学,53(4),579-590。https://doi.org/10.1111/psyp.12583
Klein Selle,N.,Verschuere,B.,Kindt,M.,Meijer,E.和Ben-Shakhar,G.(2017)。揭示定向和抑制在隐藏信息测试中的作用:CIT 中的定向和抑制。 心理生理学,54(4),628-639。https://doi.org/10.1111/psyp.12825
Rosenfeld,JP,Ozsan,I.和Ward,AC(2017)。在模拟模拟犯罪中嫌疑人与证人角色的参与者之间,Pz 的 P300 振幅和 F3 的 N200/N300 潜伏期不同:CIT 中的 P300 和 N200/N300。 心理生理学,54(4),640-648。https://doi.org/10.1111/psyp.12823

Reviewer 2 Report
Comments and Suggestions for Authors
The present study investigates event-related potentials in a Concealed Information Test. I summarize my recommendations subsequently:
The Introduction is widely concisely written. Thus, I rather have minor recommendations for the Introduction:
Line 36: I suppose the authors refer to individuals who were grouped in mock-crime paradigms into a group who had mock-crime knowledge. Thus, I recommend the authors write “mock-guilty participants” or “participants who had knowledge of a mock crime”.
Line 37: As physiological responses of participants in a mock-guilty group are usually not investigated solely I recommend the authors also mention the group of comparison (e.g., participants who did not have knowledge of a mock-crime and, thus, were named innocent).
Line 57: Because there are meta-analyses for response time data, for peripherphysiological data and for the P3 component of the event-related potential. It seems to be rather an understatement for the remarkable number CIT studies. The authors are asked to cite a representative number of studies in the filed of CIT research.
Lines 64 to 69: The fact that the P3 amplitude can be more positive to probe but also less positive depends on the type of CIT paradigm and the applied stimuli. Koeckritz et al. (2019) have shown that CIT paradigms with known stimulus sets (i.e., probe, target-1 and target-2 are all known from the practice or learning phase) evoke a mental effort effect with the probe-P3 being smaller compared to targets. CITs with known (probe, target) and unknown stimuli (irrelevant) evoke a salience effect with a more positive probe-P3 compared to irrelevant. I recommend the authors also explain these different mechanisms underlying P3 effects and also refer to the corresponding meta-analysis published in 2019 in Brain & Cognition.
Method section:
Section 2.3: Please report the company of the EEG equipment.
Please also report whether horizontal and vertical ocular electrodes were placed and where.
Please also report the algorithm for the ICA.
The filter is rather suitable for analyzing P3 components. I recommend the authors think of specific -especially high-pass filtering- for the N2 component.
Please also check whether the reported sequence: ICA followed by filtering was indeed the applied syntax sequence for EEG pre-processing. Otherwise, please report the pre-processing sequence in accordance with the syntax in Brain-Vision Analyzer.
The criterion range for correcting muscle artefacts is quite small compared to other CIT studies investigating event-related potentials. I recommend the authors might consult the P3 studies reported in two separate P3 meta-analyses on CIT paradigms for deciding about another muscle artefact rejection criterion.
In the same line, I would prefer a summary of the number of artefact-free epochs available for probe, target, and irrelevant stimuli.
The authors performed a temporal PCA. In line 195-196 the authors report a Scree plot. I recommend the authors present the Figure of the Screeplot – and at best a parallel analysis. Please also report the kappa for Promax rotation.
The PCA appears to be a temporal PCA, which is normally not applied at one electrode (Pz or Fz, Figure 4). Figure 4 is not quite clear about the different disentangled PCA components. The plot is rather untypical for components after tPCA. Please check tPCA studies and component plots This might be due to the fact that the tPCA has been performed at one electrode position. This appears to be a situation for re-newing the tPCA including all electrodes except mastoids and electrodes for controlling eye movements.
In Figure 2, I could not clearly outline a P3 component and the FNSW, respectively. Might this be due to a quite low number of artefact-free epochs (e.g., less than 10 epochs per stimulus type)? Please check and possibly revise (wider muscle artefact rejection criterion?).
Results
Please explain the expected magnitude of a BF in relation to a p-value of significance. Otherwise, it might be more difficult to understand the author’s description in line 226.
Figure 5: I also did not catch the point why the authors analyzed at 1ß22 ms and 1850 ms post-stimulus.
At least some results do not seem to refer to tPCA component scores but to other quantifications of the event-related potentials. If the authors would decide to report baseline-to-peak, mean, peak-to-peak results. Please outline the type of quantifying the ERPs and which time frame was used for quantifying event-related potentials.
Because of these questions to be clarified I refrain from giving feedback for the Discussion section.
Minor issues:
Please correct the author’s name: Rosenfeld instead of Ronsenfeld.
Line 56: should be “mechanisms underlying CIT…”
Line 59: “did” could be deleted
Line 122: “Eventually” might be deleted
Line 132: “showing” could be replaced with “showed”
Comments on the Quality of English Language
see above: minor issues
Author Response
Reviewer 2
The present study investigates event-related potentials in a Concealed Information Test. I summarize my recommendations subsequently:
The Introduction is widely concisely written. Thus, I rather have minor recommendations for the Introduction:
Response: We very much appreciate the reviewer's evaluation of our work. Your comments and suggestions have been very helpful for our research.
Comment 1: Line 36: I suppose the authors refer to individuals who were grouped in mock-crime paradigms into a group who had mock-crime knowledge. Thus, I recommend the authors write “mock-guilty participants” or “participants who had knowledge of a mock crime”.
Response: Thanks for this suggestion. We have changed "individuals" to "participants who had knowledge of a mock crime" to make the expression more exact as suggested.
"For participants who had knowledge of a mock crime, the probe is distinguishable from irrelevant stimuli and holds significance, evoking unique physiological responses. “See page 1
Comment 2: Line 37: As physiological responses of participants in a mock-guilty group are usually not investigated solely I recommend the authors also mention the group of comparison (e.g., participants who did not have knowledge of a mock-crime and, thus, were named innocent).
Response: Thanks for this valuable suggestion. We have supplemented the comparison of the differences between “participants who did not have knowledge of a mock crime” and “participants who had knowledge of a mock crime” in this section to make the expression more exact.
"In contrast, for participants who did not have knowledge of a mock crime, the probe resembles irrelevant stimuli and lacks significance, evoking physiological responses similar to irrelevant stimuli. “See page 1
Comment 3:Line 57: Because there are meta-analyses for response time data, for peripherphysiological data and for the P3 component of the event-related potential. It seems to be rather an understatement for the remarkable number CIT studies. The authors are asked to cite a representative number of studies in the filed of CIT research.
Response 3: Thanks for this suggestion. We have cited two relevant meta-analyses papers in the introduction now: Leuea & Beauducel, 2019 and Sai et al., 2023.
“P3 is a positive ERP component which typically appears 300-800ms after stimuli, and it has been found to be sensitive to stimuli that are significant and meaningful to participants (Johnson, 1986; Polich, 2007; for a meta, see Leue & Beauducel, 2019).”see page 2
“Second, given that prior research did not find an increased N2 effect in the conceal condition (Rosenfeld et al., 2017; Matsuda & Nittono, 2018; klein Selle et al., 2021; for a meta, see Sai et al., 2023), we did not expect that probe would produce a more negative N2 amplitude than irrelevant in the conceal condition. “see page 3
Comment 4:Lines 64 to 69: The fact that the P3 amplitude can be more positive to probe but also less positive depends on the type of CIT paradigm and the applied stimuli. Koeckritz et al. (2019) have shown that CIT paradigms with known stimulus sets (i.e., probe, target-1 and target-2 are all known from the practice or learning phase) evoke a mental effort effect with the probe-P3 being smaller compared to targets. CITs with known (probe, target) and unknown stimuli (irrelevant) evoke a salience effect with a more positive probe-P3 compared to irrelevant. I recommend the authors also explain these different mechanisms underlying P3 effects and also refer to the corresponding meta-analysis published in 2019 in Brain & Cognition.
Response: Thanks for this comment. We agree with the reviewer that P3 effects may reflect different mechanisms. We have added sentences to discuss the mental effort account of P3 and added relevant references.
“ It should be noted that some previous research suggests that mental effort also modulates P3 amplitudes (Koeckritz et al., 2019; Leue & Beauducel, 2019). For example, Koeckritz et al. (2019) reported that concealing untrustworthiness showed the smaller parietal mean P3 amplitude compared to truthful response towards untrustworthy faces (Koeckritz et al., 2019). Given that arousal inhibition typically requires much mental effort, thus, it is also possible that P3 is also influenced by arousal inhibition.“See page 10
Comment 5:Section 2.3: Please report the company of the EEG equipment. Please also report whether horizontal and vertical ocular electrodes were placed and where.Please also report the algorithm for the ICA.
Response: Thanks for this comment. In the methods section, we have added the follow information: the company of the EEG equipment, horizontal and vertical ocular electrodes as well as the ICA algorithm utilized.
"In this study, we utilized 32 pre-amplified Ag/AgCl electrodes and antiCAP system (Brain Products, Germany) to collect the participants’ electroencephalogram signals with placing the electrodes on the elastic electroencephalogram cap following the 10-20 system standard."see page 5
"Due to the lack of extra horizontal and vertical ocular electrodes, we selected Fp2 as the vertical eye electrode. Electrooculogram were filtered out in a semi-automatic mode and marked, with the amplitude change exceeds the average value or the voltage change value of the amplitude exceeds 97% considered as Electrooculogram. Then, remove channels with the artifact rate greater than 20% through raw data inspection."see page 5
Comment 6: The filter is rather suitable for analyzing P3 components. I recommend the authors think of specific -especially high-pass filtering- for the N2 component.
Response: Thanks for the question. PCA analyses usually require a relative large filter range so that we choose 0.1-30 Hz, which includes the frequency of both P300 and N2 (Azizian et al., 2006; Pfister et al., 2014; Rosenfeld et al., 2017; Matsuda & Nittono, 2018).
Comment 7: Please also check whether the reported sequence: ICA followed by filtering was indeed the applied syntax sequence for EEG pre-processing. Otherwise, please report the pre-processing sequence in accordance with the syntax in Brain-Vision Analyzer.
Response: Thanks for this comment. In this study, we initially employed eye movement correction using the ICA method to remove eye artifacts from the raw data, followed by using Raw Data Inspection to eliminate artifacts, removing channels with artifact rates exceeding 20%. Subsequently, digital filtering was applied to the EEG signals. We also added this in the manuscript now (see page 4).
Comment 8:The criterion range for correcting muscle artefacts is quite small compared to other CIT studies investigating event-related potentials. I recommend the authors might consult the P3 studies reported in two separate P3 meta-analyses on CIT paradigms for deciding about another muscle artefact rejection criterion.
Response: Thanks for this comment. In this study, trials with signals exceeding ±100μV were defined as artifact trials and excluded. This criterion is widely used in many previous studies. (e.g. Carrión et al., 2010; Rosenfeld et al., 2015; Fu et al., 2017)
Comment 9: In the same line, I would prefer a summary of the number of artefact-free epochs available for probe, target, and irrelevant stimuli.
Response: We thank the reviewer for this suggestion. We have added a summary of the number of artifact-free epochs available for probe and irrelevant stimuli. It reads as follow: “After correcting eye movements and removing artifacts, for participants in the conceal condition, the mean number of valid trials for probe was 34.45 (SE = 1.29) segments, while for irrelevant stimuli, it was 136.75 (SE = 5.31) segments. For participants in the reveal condition, the mean number of valid trials for detecting stimuli was 34.10 (SE = 1.19) segments, and for irrelevant stimuli, it was 139.1. (SE = 4.72) segments. “see page 5
Comment 10:The authors performed a temporal PCA. In line 204-205 the authors report a Scree plot. I recommend the authors present the Figure of the Scree plot – and at best a parallel analysis. Please also report the kappa for Promax rotation.
Response: We thank the reviewer for this suggestion. We have reported the kappa for Promax rotation in the manuscript now and added the Scree plot in supplementary materials.
“Specifically, the temporal PCA was conducted by combining data from the conceal and reveal conditions and using the Promax Rotation method with a kappa of 3, which is the parameter that determines how oblique the rotations will be(Dien et al., 2007).”see page 5
“Supplementary Materials: The Scree plot of temporal PCA. We utilized a parallel test to compare the Scree plot of the dataset with that of a completely random dataset. Adjust the scaling (on the left) to 30 to identify the point where the blue line intersects with the blue line. In this instance, 26 factors are indicated as the number (the last point above the line).”see supplementary materials.
Comment 11:The PCA appears to be a temporal PCA, which is normally not applied at one electrode (Pz or Fz, Figure 4). Figure 4 is not quite clear about the different disentangled PCA components. The plot is rather untypical for components after tPCA. Please check tPCA studies and component plots This might be due to the fact that the tPCA has been performed at one electrode position. This appears to be a situation for re-newing the tPCA including all electrodes except mastoids and electrodes for controlling eye movements.
Response: Thanks for this comment. In Temporal PCA, we didn't analyze a single electrode individually but rather conducted analysis on all 32 electrodes. Subsequently, we selected Pz and Fz for reporting based on the topographical characteristics of P3 and FNSW.
Comment 12:In Figure 2, I could not clearly outline a P3 component and the FNSW, respectively. Might this be due to a quite low number of artefact-free epochs (e.g., less than 10 epochs per stimulus type)? Please check and possibly revise (wider muscle artefact rejection criterion?).
Response: Thanks for this question. This is because sometimes we could not see PCA-based ERPs in an original ERP wave. We checked the artifact-free epochs and almost all of them are more than 30.
“in the conceal condition, the mean number of valid trials for probe was 34.45 (SE = 1.29) segments” “For participants in the reveal condition, the mean number of valid trials for detecting stimuli was 34.10 (SE = 1.19) segments”see page 5
Comment 13:Please explain the expected magnitude of a BF in relation to a p-value of significance. Otherwise, it might be more difficult to understand the author’s description in line 238.
Response: Thank you for this comment. We have explained the expected magnitude of a BF in relation to a p-value of significance in section 2.4.
“Group analyses were conducted using ANOVA methods and t-tests, with the effect size of significant findings expressed through partial eta squared and Cohen’s d. Additionally, Jeffreys–Zellner–Siow (JZS) Bayes factors (BFs) (see Rouder et al.,, 2009) were employed to supplement traditional statistical inference. The BF is a numerical value that serves as a means of quantifying the ratio of the probability of the null hypothesis to the probability of the alternative hypothesis (Kass & Raftery, 1995).”see page 5
“A BF value of ≥3 was regarded as moderate evidence for the respective hypothesis (Kass & Raftery, 1995).”see page 6
Comment 14:Figure 5: I also did not catch the point why the authors analyzed at 1022 ms and 1850 ms post-stimulus.
Response: We apologize for not being clear enough. Given that some previous research showed that a frontal negative slow wave occurs around 500 to 1000ms after presenting the stimulus, which is associated with arousal inhibition. However, we did not find a negative slow wave in that time window, but found two negativities around 1022 and 1085ms. Thus, we analyzed those two negative ERPs components for exploratory purpose. We have made this clear in page 6.
“Based on the temporal and spatial characteristics of P3, two factors were chosen for analyses. These factors included a positivity peaking at 400 at Pz, and a positivity peaking at 590ms at Pz. We did not find any components that were temporally and spatially similar to N2, or FNSW. Nonetheless, we observed a negativity peak at 1022ms at Fz, and a negativity peak at 1850ms at Fz. These were examined as exploratory analyses.”
Comment 15:At least some results do not seem to refer to tPCA component scores but to other quantifications of the event-related potentials. If the authors would decide to report baseline-to-peak, mean, peak-to-peak results. Please outline the type of quantifying the ERPs and which time frame was used for quantifying event-related potentials.
Response: Thanks for this suggestion. We indeed used PCA to score all the components we focused on. Thus, we did not report baseline-to-peak, mean, peak-to-peak results and time frame was used for quantifying event-related potentials.
Comment 16:Please correct the author’s name: Rosenfeld instead of Ronsenfeld. Line 56: should be “mechanisms underlying CIT…”. Line 59: “did” could be deleted. Line 122: “Eventually” might be deleted. Line 132: “showing” could be replaced with “showed”
Response: Thanks for these suggestions. We have changed them in the manuscript now.
References
Azizian, A., Freitas, A. L., Parvaz, M. A., & Squires, N. K. (2006). Beware misleading cues: perceptual similarity modulates the N2/P3 complex. Psychophysiology, 43(3), 253–260. https://doi.org/10.1111/j.1469-8986.2006.00409.x
Carrión, R. E., Keenan, J. P., & Sebanz, N. (2010). A truth that’s told with bad intent: An ERP study of deception. Cognition, 114(1), 105–110. https://doi.org/10.1016/j.cognition.2009.05.014
Dien, J., Khoe, W., & Mangun, G. R. (2007). Evaluation of PCA and ICA of simulated ERPs: Promax vs. infomax rotations. Human Brain Mapping, 28(8), 742–763. https://doi.org/10.1002/hbm.20304
Fu, H., Qiu, W., Ma, H., & Ma, Q. (2017). Neurocognitive mechanisms underlying deceptive hazard evaluation: An event-related potentials investigation. PloS one, 12(8), e0182892. https://doi.org/10.1371/journal.pone.0182892
Kass, R. E., & Raftery, A. E. (1995). Bayes Factors. Journal of the American Statistical Association, 90(430), 773–795. https://doi.org/10.1080/01621459.1995.10476572
Koeckritz, R., Beauducel, A., Hundhausen, J., Redolfi, A., & Leue, A. (2019). Does concealing familiarity evoke other processes than concealing untrustworthiness? – Different forms of concealed information modulate P3 effects. Personality Neuroscience, 2. https://doi.org/10.1017/pen.2019.4
Leue, A., & Beauducel, A. (2019). A meta-analysis of the P3 amplitude in tasks requiring deception in legal and social contexts. Brain and cognition, 135, 103564. https://doi.org/10.1016/j.bandc.2019.05.002
Matsuda, I., & Nittono, H. (2018). A concealment-specific frontal negative slow wave is generated from the right prefrontal cortex in the Concealed Information Test. Biological Psychology, 135, 194–203. https://doi.org/10.1016/j.biopsycho.2018.04.002
Pfister, R., Foerster, A., & Kunde, W. (2014). Pants on fire: the electrophysiological signature of telling a lie. Social neuroscience, 9(6), 562–572. https://doi.org/10.1080/17470919.2014.934392
Rosenfeld, J. P., Ozsan, I., & Ward, A. C. (2017). P300 amplitude at Pz and N200/N300 latency at F3 differ between participants simulating suspect versus witness roles in a mock crime: P300 and N200/N300 in a CIT. Psychophysiology, 54(4), 640–648. https://doi.org/10.1111/psyp.12823
Rosenfeld, J. P., Ward, A., Frigo, V., Drapekin, J., & Labkovsky, E. (2015). Evidence suggesting superiority of visual (verbal) vs. auditory test presentation modality in the P300-based, Complex Trial Protocol for concealed autobiographical memory detection. International Journal of Psychophysiology, 96(1), 16–22. https://doi.org/10.1016/j.ijpsycho.2015.02.026
Rouder, J. N., Speckman, P. L., Sun, D., Morey, R. D., & Iverson, G. (2009). Bayesian t tests for accepting and rejecting the null hypothesis. Psychonomic Bulletin & Review, 16(2), 225–237. https://doi.org/10.3758/PBR.16.2.225
Sai, L., Cheng, J., Shang, S., Fu, G., & Verschuere, B. (2023). Does deception involve more cognitive control than truth‐telling? Meta‐analyses of N2 and MFN ERP studies. Psychophysiology, 60(10), e14333. https://doi.org/10.1111/psyp.14333

Round 2
Reviewer 2 Report
Comments and Suggestions for Authors
The authors addressed several of my recommendations. Thanks.
However, the main concern for calculating and presenting the tPCA components remains: Based on Figure 4, the tPCA is still inconclusive compared with other published tPCA results. Readers would expect to find several tPCA components for the chosen time range (x-axis) in the Figure. The tPCA should have been performed across all EEG electrodes -except mastoids and occular electrodes- because authors present a time-related PCA, no spatial PCA.
